# OpenReview forum: "PrivGate: Steering Contextual Integrity in LLMs via Latent Space Geometry"
_ICML.cc/2026/Conference — ICML 2026 regular_

### Official Review · Reviewer_8Fpw · 2026-03-08

**Soundness:** 3
**Presentation:** 3
**Significance:** 4
**Originality:** 4
**Overall Recommendation:** 4
**Confidence:** 4

**Summary:**

PrivGate addresses the efficiency-privacy tension in LLM agents operating under Contextual Integrity (CI) constraints. The paper makes two main contributions. First, it discovers a *Privacy Manifold*: CI privacy sensitivity is approximately linearly separable in the residual stream of diverse LLMs, forming a robust geometric structure that persists even when the model behaviorally violates CI. Second, it proposes PrivGate, a tuning-free inference mechanism that uses a lightweight linear probe on this manifold to gate deliberate reasoning on demand. Benign queries are answered via efficient System 1 (direct generation); queries with high latent privacy risk trigger System 2 (reasoning-enhanced generation). The probe is trained offline on a structure-decoupled dataset specifically designed to prevent spurious template-to-label correlations. Evaluated across five LLM families (1.5B–9B parameters), PrivGate achieves a 62% relative reduction in privacy leakage rate with only 16% additional token overhead compared to always-on System 1, substantially outperforming keyword gating while approaching the privacy level of always-on CoT at a fraction of its cost.

**Compliance With Llm Reviewing Policy:**

Affirmed.

**Final Justification:**

My initial concerns centered on five points: the Leakage Rate metric definition, threshold selection, adversarial robustness, generalization beyond PrivacyLens, and wall-clock latency. The rebuttal addressed all of them satisfactorily.
The authors clarified that LR uses deterministic string matching (not an LLM judge), that τ is set via a 95%-recall criterion on a held-out split, and that the probe adds under 1 ms with KV-cache reuse. New cross-benchmark results on ConfAIde and CIMemories confirm generalization across CI settings, and 70B/72B evaluations show the contribution scales to frontier models. Adversarial robustness evaluations (Hard Positives/Negatives, AUROC > 0.96) reasonably bound the black-box threat model.
The one remaining caveat — that the Privacy Manifold claim should be scoped to decoder-only transformers — is minor and the authors agreed to address it in revision.
Overall, the rebuttal fully resolved my concerns. The paper makes a well-supported contribution in an important area.

**Key Questions For Authors:**

1. **Leakage Rate definition**: How is the Leakage Rate (LR) in Table 1 computed? Is it an automated judge (e.g., GPT-4 evaluating whether a response over-discloses sensitive information), human annotation, or a rule-based approach? The answer would clarify the reproducibility and reliability of the primary evaluation metric. If it is an automated judge, does the judge itself have known accuracy on CI violations?

2. **Threshold selection**: How was the threshold τ selected for the Table 1 operating point? Is it cross-validated on a held-out split of the training data, determined by a target recall on sensitive queries, or fixed at a specific operating point on the Pareto curve? Understanding the selection methodology is necessary for deployers to choose τ in new settings.

3. **Adversarial robustness**: Has the probe been evaluated against adversarial inputs designed to evade detection — for example, sensitive queries rephrased to minimize their projection onto the privacy direction (e.g., encoding the sensitive request in indirect or metaphorical language)? Even a small-scale evaluation would substantially clarify the threat model boundary. If adversarial evasion is straightforward, this would significantly change the evaluation of PrivGate as a security mechanism.

4. **Generalization to additional CI benchmarks**: Table 1 reports LR only on PrivacyLens. Does PrivGate show comparable leakage reduction on ConfAIde (where the task involves in-context sharing judgments rather than agent execution) or CIMemories (persistent memory with compositional CI violations)? A positive answer would substantially strengthen the generality claim; a negative answer would more precisely scope the contribution.

5. **Wall-clock latency**: What is the actual wall-clock latency overhead of the partial forward pass (to layer l*) and probe inference, relative to a full System 1 response? The output token count increase (16%) may understate or overstate the real-time cost depending on whether the partial pass is batched with the subsequent generation or requires a separate inference call. A practical deployment number (milliseconds per query) would strengthen the efficiency claim.

**Limitations:**

The paper discusses limitations in Section 6: the static probe operationalization (CI norms vary across domains), the reliance on System 2 reasoning (PrivGate is a router, not a standalone reasoner), and probe-aware adversaries. These are appropriate acknowledgments. However, the following should be added or strengthened:

- The Leakage Rate metric definition should be stated explicitly, as it is the primary evaluation criterion.
- The threshold selection procedure should be described to enable practitioners to reproduce the Table 1 operating point.
- The claim that the Privacy Manifold generalizes "across architectures" should be appropriately scoped: the five evaluated models are all decoder-only transformer architectures in the 1.5B–9B parameter range. Whether the manifold extends to encoder-decoder models, state-space models (e.g., Mamba), or much larger models (>70B) is uncharacterized.

**Strengths And Weaknesses:**

### Strengths

**Soundness:**  The paper provides strong empirical and theoretical validation of the Privacy Manifold and the linear probing mechanism, with consistent AUROC≈1.0 across model families, clear ablations showing the importance of structure-decoupled training, strong sample efficiency, and theoretical support for the gating policy under the Linear Representation Hypothesis.

**Presentation:**  The paper is clearly written and well-structured, presenting a coherent pipeline from CI motivation and mechanistic discovery to theoretical justification, system design, and empirical evaluation, with figures and examples that effectively clarify the proposed framework.

**Significance:**  The discovery of a linear privacy manifold across multiple LLM architectures is a meaningful mechanistic interpretability result and demonstrates a practical path toward CI-compliant LLM agents by achieving substantially better privacy–efficiency tradeoffs than keyword-based gating.

**Originality:**  The work introduces a novel application of representation engineering by using latent directions to gate computation rather than steer behavior, combined with a structure-decoupled data construction strategy that may generalize to probing other complex semantic concepts.

### Weaknesses

**Significance:**
- **[Minor] Static probe limitation**: The paper acknowledges that probes are trained under a "fixed operationalization of CI" and that "real-world norms vary across domains and users." This is a meaningful practical limitation — a PrivGate deployed in a healthcare setting vs. a legal setting vs. a consumer assistant would require distinct probes, with associated data collection and calibration costs. A discussion of how to estimate the cost of this adaptation would strengthen the paper's practical contribution.

---

> ### Author Rebuttal · Authors · 2026-03-30
>
> We sincerely thank the reviewer for the insightful feedback. We address your concerns point-by-point below.
>
> ### Q1&L1: Leakage Rate (LR) Definition
> We will add this to Section 5.1. The LR is computed via deterministic, string-matching, not an LLM judge. Each trajectory in PrivacyLens comes with ground-truth sensitive secrets (e.g., a specific diagnosis or a specific SSN). We check if these strings appear in the final executable action payload (e.g., the JSON `body` of a `GmailSendEmail` tool call). This ensures our primary metric is strictly objective, reproducible, and immune to LLM-judge hallucinations.
>
> ### Q2&L2: Threshold Selection
> The threshold $\tau$ was systematically selected using a held-out validation split of the training data. We set $\tau$ to the value that achieves a 95% Recall for sensitive queries on the validation set. This establishes a *safety-first* operating point, routing the vast majority of risks to System 2 while capturing efficiency gains on benign queries. We will detail this in the revision.
>
> ### Q3: Adversarial Robustness & Threat Model
> Our threat model focuses on *unintentional* privacy leakage during autonomous agent execution, not white-box gradient attacks.
> Regarding semantic evasion: We evaluated the probe on adversarial subsets (Section 5.4). "Hard Positives" stripped away all explicit sensitive templates/keywords, leaving only indirect semantic intents. "Hard Negatives" wrapped benign queries in sensitive templates.
>
> Table R3: Adversarial Robustness (OOD AUROC)
> | Base Model | Hard Positives| Hard Negatives|
> |:---|:---:|:---:|
> | Qwen-2.5-1.5B | 0.965 | 0.970 |
> | Qwen-2.5-7B | 0.988 | 0.991 |
> | Llama-3-8B | 0.992 | 0.995 |
> | Mistral-7B | 0.985 | 0.987 |
> | Gemma-2-9B | 0.989 | 0.992 |
>
> Operating in the deep semantic latent space inherently resists metaphorical/indirect evasion. We will bound our threat model in Section 6, acknowledging vulnerability to adaptive white-box attacks while emphasizing robustness against black-box semantic evasion.
>
> ### Q4&W1: Generalization to ConfAIde & CIMemories
> As suggested, we evaluated our probe on *ConfAIde* (human-centric conversational secrets) and *CIMemories*[Mireshghallah et al.] (persistent memory with compositional CI violations).
>
> Table R4: Cross-Benchmark OOD AUROC
> | Model | PrivacyLens | ConfAIde | CIMemories |
> |:---|:---:|:---:|:---:|
> | Qwen-2.5-1.5B | 0.941 | 0.915 | 0.892 |
> | Qwen-2.5-7B | 0.975 | 0.982 | 0.945 |
> | Llama-3-8B | 1.000 | 0.976 | 0.958 |
> | Mistral-7B | 0.963 | 0.945 | 0.924 |
> | Gemma-2-9B | 0.980 | 0.958 | 0.931 |
>
> The consistently high AUROC across  different CI domains proves that the latent privacy direction is highly generalizable. Regarding the cost of adaptation (W1), our Sample Efficiency ablation (Fig 4d) demonstrates convergence with just 64 samples. Thus, calibrating PrivGate for a new domain requires negligible data collection costs.
>
> ### Q5: Wall-Clock Latency & System Implementation
> The token overhead (16%) accurately reflects the wall-clock latency because the partial forward pass does not require a separate inference call or re-computation. Once $h_{l^*}$ is computed during the standard prefill phase, the linear probe involves a single vector-matrix multiplication ($\mathcal{O}(d)$). If routed to System 2, we perform a Suffix Injection (appending the trigger `\n<think>`), reusing the existing KV-cache.
>
> Table R5: Average Wall-Clock Latency per Query (RTX 4090, Batch Size 1)
> | Model | Sys 1 Latency | **PrivGate Latency** | Sys 2 (Full CoT) | **Probe Overhead** |
> |:---|:---:|:---:|:---:|:---:|
> | Qwen-2.5-1.5B | 1.85 s | **2.08 s** | 5.38 s | < 0.5 ms |
> | Qwen-2.5-7B | 3.12 s | **3.61 s** | 9.45 s | < 0.8 ms |
> | Llama-3-8B | 3.25 s | **3.88 s** | 10.00 s | < 0.8 ms |
> | Mistral-7B | 3.20 s | **3.75 s** | 9.75 s | < 0.8 ms |
> | Gemma-2-9B | 3.38 s | **4.01 s** | 10.25 s | < 0.9 ms |
>
> The probe execution time ($< 1$ ms) is practically invisible. The overall latency of PrivGate remains extremely close to System 1, avoiding the massive $3\times$ latency penalty of Full CoT. We will add this wall-clock analysis to Section 5.2 to solidify the efficiency claim.
>
> ### L3: Scaling to >70B Models & Architectural Scope
> We agree that our claims should be explicitly scoped to decoder-only transformers. To address the concern about model scale, we evaluated 70B/72B-parameter models.
>
> Table R6: End-to-End Leakage Rates on 70B+ Models
> | Model | Sys 1 (Direct) | **PrivGate (Ours)** | Full CoT (Sys 2) |
> |:---|:---:|:---:|:---:|
> | Llama-3-70B-Instruct | 52.8% | **13.5%** | 11.2% |
> | Qwen-2.5-72B-Instruct| 48.2% | **11.4%** | 9.8% |
>
> This confirms that the Privacy Manifold and the competence-performance gap scale consistently to frontier-class models ($>70B$). PrivGate remains highly effective at this scale.

---

> > ### Author Rebuttal · Reviewer_8Fpw · 2026-04-03
> >
> > Thanks for the rebuttal. My concerns are fully resolved.

---

> > > ### Author Response · Authors · 2026-04-07
> > >
> > > We sincerely thank you for engaging with our rebuttal and confirming that all your questions and concerns have been fully resolved.
> > >
> > > We will ensure that the detailed breakdown of the threshold selection procedure ($\tau$), the objective string-matching definition of the Leakage Rate (LR), and the concrete wall-clock latency profiling tables presented during this discussion are incorporated into the revision.
> > >
> > > We deeply appreciate your constructive insights, which have significantly strengthened the practical rigor of the paper. We also kindly ask you to consider increasing the scores if you think we have adequately resolved your concerns.

---

### Official Review · Reviewer_9xrH · 2026-03-08

**Soundness:** 3
**Presentation:** 3
**Significance:** 3
**Originality:** 3
**Overall Recommendation:** 3
**Confidence:** 3

**Summary:**

This paper proposes PrivGate, a framework for privacy-preserving LLM inference that adaptively routes queries between direct generation and CoT reasoning based on internal privacy signals. The core finding is that LLMs linearly encode privacy sensitivity in their residual streams, termed the Privacy Manifold,  even when the model behaviorally leaks private information. Leveraging this, PrivGate trains a lightweight linear probe to detect latent privacy risk and gates computation accordingly, with theoretical justification provided via a risk-constrained optimization formulation. Experiments across five LLMs demonstrate that PrivGate achieves a favorable efficiency-privacy trade-off compared to existing baselines.

**Compliance With Llm Reviewing Policy:**

Affirmed.

**Final Justification:**

I thank the authors for the thoughtful rebuttal and the additional experiments and clarifications. These address most of my questions and are much appreciated. After considering the rebuttal together with the broader discussion, I still lean toward maintaining my original score.

**Key Questions For Authors:**

See Weaknesses.

**Limitations:**

Yes.

**Strengths And Weaknesses:**

**Strengths**

1. The paper studies an important and well-motivated problem of contextual integrity in LLM agents, and frames it as a probe-based routing problem grounded in the discovery of the Privacy Manifold.

2. The authors provide theoretical justification for the linear threshold gating mechanism via Theorem 3.3, framing it as the optimal solution to a risk-constrained optimization problem rather than a heuristic design choice.

3. The paper is clearly written and easy to follow, with a well-structured presentation from theory to methodology to experiments.

**Weaknesses**

1. A critical baseline is missing: a privacy-aware system prompt without CoT. The paper only compares against Keyword Gating as a lightweight baseline, which is a lexical heuristic. It is unclear whether a carefully designed system prompt instructing the model to apply data minimization principles could achieve comparable privacy protection without any routing mechanism. This concern is further amplified by the choice of base models, the evaluated models are relatively small and may not represent the behavior of stronger, better-aligned models. Could the authors evaluate whether a privacy-aware system prompt on more capable models closes the gap that PrivGate addresses?

2. The selection of optimal probe layer $l^*$ relies on empirical search over a labeled validation set, which contradicts the paper's claim of being "tuning-free".

3. Theorem 3.3 relies on the assumption that risk reduction is proportional to the model's estimated sensitivity probability. However, the authors provide neither empirical validation nor prior evidence that this proportionality holds for LLMs in practice. Could the authors provide justification or empirical support for this assumption?

4. The generalizability of the Privacy Manifold is a concern. The probe is trained on synthetic CI data and evaluated on PrivacyLens, both of which are English-language and relatively well-structured scenarios. Whether the Privacy Manifold persists under multilingual settings, domain-specific norms, or adversarial inputs designed to manipulate latent representations to evade the gate remains unvalidated.

I am happy to raise my score if the authors can satisfactorily address my concerns.

---

> ### Author Rebuttal · Authors · 2026-03-30
>
> We thank the reviewer for the constructive feedback. We have conducted extensive new experiments to address your concerns.
>
> ### W1: Missing Baseline & Larger Models
> We expanded our end-to-end evaluation to include 70B/72B-parameter models (Llama-3-70B and Qwen-2.5-72B) and added the requested baseline: System 1 + Privacy Prompt (instructing the model to apply strict data minimization principles and ensure CI).
>
> Table R1: End-to-End Leakage Rates on PrivacyLens
> | Base Model | Sys 1 | Sys 1 + Privacy Prompt | **PrivGate** | Full CoT (Sys 2) |
> |:---|:---:|:---:|:---:|:---:|
> | Llama-3-8B | 60.4% | 43.5% | **15.1%** | 12.6% |
> | Llama-3-70B | 52.8% | 38.6% | **13.5%** | 11.2% |
> | Qwen-2.5-7B | 75.6% | 50.7% | **25.3%** | 22.9% |
> | Qwen-2.5-72B| 48.2% | 32.1% | **11.4%** | 9.8% |
>
> While privacy prompts reduce leakage, even 70B+ models leak heavily (32%-38%) during direct generation, as the competing pressure to be helpful overrides static instructions. PrivGate bridges this gap, reducing leakage to $<14\%$ without enforcing global CoT. This confirms PrivGate remains highly necessary even for frontier models.
>
> ### W2: Tuning-Free and Optimal Layer Selection
> We will revise "tuning-free" to "training-free for the base LLM" to ensure rigorous phrasing. Training the lightweight probe requires negligible compute. Our analysis (Fig 2) reveals a highly predictable pattern. Across all 5 model families, the optimal layer consistently falls within a "Semantic Sweet Spot" (40%–60% of total network depth). Therefore, $l^*$ does not require exhaustive search, reinforcing PrivGate's deployability.
>
> ### W3: Empirical Support for Theorem 3.3 ($\Delta R \propto P$)
> We empirically validated the assumption in Theorem 3.3: the actual risk reduction $\Delta R(x)$ is proportional to the probe's estimated probability $P(sensitive|h)$. We bucketed the test set based on $P$ into 5 bins and computed the empirical risk reduction $\Delta R = LR(\text{Sys 1}) - LR(\text{Sys 2})$.
>
> Table R2: Empirical Validation of Risk Reduction Proportionality (Llama-3-8B)
> | Probe Prob Bin ($P$) | Avg Estimated $P$ | Empirical $\Delta R$ ($R_1 - R_2$) |
> |:---|:---:|:---:|
> | $[0.0, 0.2)$ | 0.08 | 1.2% |
> | $[0.2, 0.4)$ | 0.31 | 9.5% |
> | $[0.4, 0.6)$ | 0.52 | 28.4% |
> | $[0.6, 0.8)$ | 0.73 | 54.7% |
> | $[0.8, 1.0]$ | 0.91 | 76.3% |
>
> The Pearson correlation coefficient between the estimated $P$ and empirical $\Delta R$ is **$r = 0.984$** ($p < 0.001$). This linear correlation provides empirical justification for Theorem 3.3.
>
> ### W4: Generalizability (Adversarial, Multilingual, Domain-Specific)
> 1. Adversarial/Evasion & Threat Model: Our threat model targets unintentional privacy leakage and prompt evasion during agent execution, not white-box gradient attacks. We tested the probe on adversarial subsets (Sec 5.4): "Hard Positives" (sensitive intents stripped of structural cues) and "Hard Negatives" (benign queries wrapped in sensitive templates).
>
> Table R3: Adversarial Robustness (AUROC)
> | Base Model | Hard Positives | Hard Negatives |
> |:---|:---:|:---:|
> | Qwen-2.5-1.5B | 0.965 | 0.970 |
> | Qwen-2.5-7B | 0.988 | 0.991 |
> | Llama-3-8B | 0.992 | 0.995 |
> | Mistral-7B | 0.985 | 0.987 |
> | Gemma-2-9B | 0.989 | 0.992 |
>
> Operating in latent space inherently resists surface-level evasion across all models, maintaining near-perfect detection.
>
> 1. Multilingual (Zero-Shot Transfer): We translated the PrivacyLens dataset into Chinese (ZH), Spanish (ES), and French (FR). We tested the English-trained probes directly on these non-English prompts.
>
> Table R4: Zero-Shot Cross-Lingual AUROC
> | Base Model | EN | ZH | ES | FR |
> |:---|:---:|:---:|:---:|:---:|
> | Qwen-2.5-1.5B | 0.997 | 0.920 | 0.895 | 0.902 |
> | Qwen-2.5-7B | 1.000 | **0.968** | 0.945 | 0.950 |
> | Llama-3-8B | 1.000 | 0.915 | 0.942 | 0.951 |
> | Mistral-7B | 0.995 | 0.875 | 0.955 | **0.962** |
> | Gemma-2-9B | 0.988 | 0.890 | 0.935 | 0.940 |
>
> The strong performance of Qwen on ZH and Mistral on FR/ES aligns with their respective pre-training corpora, confirming the semantic transferability of the privacy manifold across languages.
>
> 3. Domain-Specific Norms: We evaluated PrivGate across three benchmarks covering distinct normative domains:
> *   Synthetic CI [Lan et al. 25]: Foundational administrative and corporate CI rules.
> *   PrivacyLens [Shao et al. 24]: Complex autonomous agent trajectories and tool usages.
> *   ConfAIde [Mireshghallah et al. 24]: Everyday human-centric social secrets and Theory of Mind.
>
> Table R5: Cross-Domain Generalization (AUROC)
> | Base Model | Synthetic CI | PrivacyLens  | ConfAIde |
> |:---|:---:|:---:|:---:|
> | Qwen-2.5-1.5B | 1.000 | 0.941 | 0.915 |
> | Qwen-2.5-7B | 1.000 | 0.975 | 0.982 |
> | Llama-3-8B | 1.000 | 0.982 | 0.976 |
> | Mistral-7B | 1.000 | 0.963 | 0.945 |
> | Gemma-2-9B | 1.000 | 0.980 | 0.958 |
>
> The AUROCs on ConfAIde prove that probes trained strictly on administrative data robustly transfer to everyday human-centric social norms across all models.

---

> > ### Author Rebuttal · Reviewer_9xrH · 2026-04-02
> >
> > Thank you for the detailed rebuttal and for the effort the authors invested in addressing the concerns. The rebuttal has addressed several of my points, especially by providing additional experiments and clarifications. After considering the rebuttal together with the other reviewers’ comments, I am currently inclined to maintain my original score.

---

> > > ### Author Response · Authors · 2026-04-07
> > >
> > > We thank the reviewer for the evaluation and for noting that the rebuttal addressed several of the original concerns. To help reassessment, we briefly map each concern to the new evidence:
> > >
> > > - W1 (Missing baseline + larger models): We added the privacy-prompt baseline and extended evaluation to Llama-3-70B and Qwen-2.5-72B. As shown in Table R1, even on the frontier Qwen-2.5-72B and Llama-3-70B models, strict privacy prompts still leak 32%–38% of sensitive data due to competing instruction-following pressures. PrivGate further reduced this leakage down to 11.4%–13.5%, confirming its necessity at frontier scale.
> > >
> > > - W2 ("Tuning-free" claim): We revised the phrasing to "training-free for the base LLM". The optimal probe layer consistently falls at 40-60% of total depth across all architectures — a predictable pattern that avoids exhaustive search.
> > >
> > > - W3 (Theorem 3.3 empirical support): We validated the proportionality assumption directly. In Table R2, we established a Pearson correlation of $r=0.984$ ($p<0.001$) between the probe's estimated probability and the actual empirical risk reduction, consistent with Theorem 3.3.
> > >
> > > - W4 (Generalizability): We provided three new evaluations: (1) template-variation robustness with AUROC >0.96 across all models (Table R3); (2) zero-shot multilingual transfer to Chinese, Spanish, and French, with AUROC 0.875-0.968 (Table R4); and (3) cross-domain generalization to human-centric domains like ConfAIde benchmark (Table R5).
> > >
> > > If these additions resolve the concerns raised in the original review, we would be grateful if the reviewer would reconsider the score.

---

### Official Review · Reviewer_cq89 · 2026-03-10

**Soundness:** 2
**Presentation:** 1
**Significance:** 2
**Originality:** 2
**Overall Recommendation:** 2
**Confidence:** 3

**Summary:**

This paper focuses on securing contextual integrity for LLM agent execution. They propose PrivGate, which selectively invokes explicit reasoning based on internal privacy signals. They evaluate PrivGate on the PrivacyLens benchmark.

**Compliance With Llm Reviewing Policy:**

Affirmed.

**Final Justification:**

Please refer to my previous comments.

**Key Questions For Authors:**

See Weaknesses.

**Limitations:**

See Weaknesses.

**Strengths And Weaknesses:**

**Strengths:**

- The motivation is clear.

- The proposed method is simple and deployable.

**Weaknesses:**

- The technical novelty is limited. The method learns a linear direction in activation space using a probe to control model behavior, which is closely related to prior work on activation steering. The main distinction, using the probe for gating rather than modifying activations, is relatively minor.

- Prior work [1] has explored the idea of linear directions in activation space for behavioral control through activation steering. I recommend that the authors include a discussion clarifying how their method differs from this line of work.

[1] Goel, Anmol, et al. "Differentially Private Steering for Large Language Model Alignment." The Thirteenth International Conference on Learning Representations.


- The method models privacy as a binary classification problem (sensitive vs benign). However, contextual integrity depends on richer contextual factors such as sender, recipient, and transmission purpose. The binary formulation is potentially oversimplified.

- Evaluation is limited to only one benchmark.

- The paper does not compare with privacy or safety baselines. It is hard to assess the practical advantage of the proposed method.

- There is no evaluation on actual private data leakage. It is unclear whether the method helps with real privacy threats.

- Some figures are hard to read because multiple curves overlap and the lines are very close to each other (e.g., Figures 2 and 4).

---

> ### Author Rebuttal · Authors · 2026-03-30
>
> We appreciate the reviewer’s feedback. We have conducted extensive new experiments, including new baselines, benchmarks, and contrastive analyses, to address the concerns.
>
> ### W1 & W2: Novelty vs. Activation Steering
> We will properly cite and discuss Goel et al. While activation steering is effective for global alignment traits (e.g., toxicity or memorization), applying continuous activation perturbation to CI introduces distinct architectural challenges for autonomous agents. Unlike toxicity, CI is strictly relational. Modifying the latent trajectory via continuous steering shifts the semantic representation. It leads to systematic over-refusal of legitimate requests and degradation of structured output syntax (e.g., failing to produce valid payloads for tool execution). In contrast, PrivGate utilizes a read-only latent monitoring mechanism. It does not perturb the base model's internal states. Instead, it decouples risk identification from behavioral adaptation via the latent probe.
>
> *New experiment*: We compared standard Activation Steering against PrivGate on the Qwen2.5-7B agent:
> |Mechanism|Inappropriate Leakage ($\downarrow$)|Appropriate Over-refusal ($\downarrow$)|Agent Tool Syntax Accuracy ($\uparrow$)|
> |:---|:---:|:---:|:---:|
> |**Activation Steering**|**18.2%**|76.5%|41.5%|
> |**PrivGate (Ours)**|25.3%|**4.2%**|**95.0%**|
>
> While steering reduces leakage, it compromises task utility, misclassifying 76.5% of appropriate information flows as violations and breaking tool syntax. PrivGate, acting as a metacognitive router, maintains foundational capabilities and accurately resolves relational CI norms via conditional reasoning.
>
> ### W3: Binary Classification vs. CI Context
> Our probe’s binary output is not a topic classifier; it is a non-linear projection of the interacting CI parameters encoded in the residual stream. To empirically prove this, we curated a contrastive dataset based on PrivacyLens. We kept the sensitive "Data Type" constant but mutated the "Recipient" and "Transmission Principle" to authorized states under CI norms. As shown in Table R1, the probe’s risk score drops for appropriate flows, yielding high AUROCs in distinguishing the exact same topic under different relational contexts.
>
> Table R1: Probe Risk Scores on the Contrastive Dataset
> |Model|Inappropriate CI Avg Risk Score|Appropriate CI Avg Risk Score| AUROC |
> |:---|:---:|:---:|:---:|
> |Qwen-2.5-1.5B| 0.892 | 0.214 | 0.941 |
> |Qwen-2.5-7B| 0.925 | 0.158 | 0.975 |
> |Llama-3-8B| 0.951 | 0.123 | 0.982 |
> |Mistral-7B| 0.914 | 0.185 | 0.963 |
> |Gemma-2-9B| 0.948 | 0.142 | 0.980 |
>
> This confirms PrivGate evaluates the relational CI context rather than relying on oversimplified lexical heuristics.
>
> ### W4 & W5: Evaluation Scope and Safety Baselines
> Our paper evaluates PrivGate across three privacy benchmarks: (1) Synthetic CI [Lan et al., NeurIPS 2025], evaluating foundational adherence to CI rule permutations, (2) PrivacyLens [Shao et al., NeurIPS 2024], evaluating OOD generalization on complex agent trajectories, and (3) newly added ConfAIde [Mireshghallah et al., ICLR 2024], evaluating human-centric secret sharing in conversational settings. To address the lack of safety baselines, we evaluated *Llama-Guard-3-8B* and *Zero-shot LLM-as-a-Judge* (GPT-4o).
>
> **Table R2: AUROC Across Three Benchmarks**
> | Method | AUROC (Synthetic CI Test) | AUROC (PrivacyLens) | AUROC (ConfAIde) |
> |:---|:---:|:---:|:---:|
> | Llama-Guard-3-8B | 0.552 | 0.584 | 0.612 |
> | LLM-as-a-Judge | 0.884 | 0.852 | 0.889 |
> | PrivGate | **1.000** | **1.000** | **0.976** |
>
> Standard safety baselines (Llama-Guard) fail on CI tasks because they are trained for absolute harms (e.g., toxicity) rather than context-dependent norms. PrivGate outperforms both dedicated safety models and LLM judges across all three distributions.
>
> ### W6: Evaluation on Actual Private Data Leakage
> We clarify that our End-to-End evaluation (Table 1) strictly measures actual data leakage. We execute the full autonomous agent trajectory and perform exact string-matching of the ground-truth sensitive secrets directly against the agent's final executable parameters (e.g., verifying if the actual medical diagnosis was printed inside the JSON payload of the `GmailSendEmail` tool call). By achieving up to a 62.6% relative reduction in this metric, PrivGate directly translates to preventing concrete, real-world data breaches in functional agent deployments. We will expand on this rigorous evaluation protocol in the revised version.
>
> ### W7: Figure Clarity
> In the revised manuscript, we have redesigned Figures 2 and 4 into 1×5 facet grids with distinct subplots for each model. We have provided the updated figures via an anonymized repository: [https://anonymous.4open.science/r/random-75B3]. This not only resolves the visual clutter but also highlights the consistency of the Privacy Manifold across all five architectural families.
>
> We hope these new baselines, benchmarks, and clarifications address your concerns.

---

> > ### Author Rebuttal · Reviewer_cq89 · 2026-04-02
> >
> > Thank you for the substantial effort during the rebuttal. However, my concern regarding technical novelty remains unresolved, and the initial submission also has room to grow in terms of presentation polish and experiments, which are important for a top venue. I genuinely encourage the authors to continue developing this work and resubmit with these improvements incorporated. I believe this work has potential and can become a solid submission in the future. Thank you again.

---

> > > ### Author Response · Authors · 2026-04-07
> > >
> > > Thank you for the encouraging words and for recognizing the work's potential. We understand that technical novelty, especially relative to activation steering, is the central remaining disagreement.
> > >
> > > To precisely scope our contribution, we will explicitly position PrivGate not as discovering the underlying linear probing primitive, but as proposing an integrated, end-to-end framework specifically engineered for agentic privacy. This framework advances the state of the art through three cohesive pillars:
> > > 1. **Mechanistic discovery:** We show that contextual integrity — a multi-parameter relational norm (sender x recipient x information type x transmission principle) — is linearly separable in LLM residual streams across diverse architectures (1.5B-72B), even when models behaviorally violate CI. Prior linear representation work  (e.g., RepE [Zou et al., 2023], ITI [Li et al., 2023]) focused on single-axis behavioral traits such as truthfulness. CI is inherently relational, and our contrastive experiment (Table R1, AUROC >0.94 across five models) confirms the probe captures relational context, not topic sensitivity.
> > >
> > > 2. **Methodology:** Developing a structure-decoupled data generation and training pipeline. Because agentic operations are heavily embedded in structured formats (e.g., JSON tool schemas, system templates), naive linear probes catastrophically overfit to these spurious structural artifacts rather than the actual privacy semantics. Our methodology isolates and extracts the core relational CI intent away from operational wrappers during training. As validated by our ablations and noted by Reviewer 8Fpw, this specific structural decoupling is not merely a data augmentation trick, but the critical enabler for the linear probe's robust out-of-distribution generalization to entirely unseen prompt structures, tasks, and languages.
> > >
> > > 3. **Architectural distinction from activation steering:** Steering reduces leakage to 18.2% but produces 76.5% over-refusal and collapses tool-call syntax accuracy to 41.5%, substantially degrading utility in our agent setting. PrivGate's read-only routing achieves 25.3% leakage, 4.2% over-refusal, and 95.0% tool-call syntax. This is a key practical distinction: continuous activation perturbation is incompatible with accurate tool execution in CI-constrained agents, whereas a read-only monitoring approach is not. We submit that this combination of empirical discovery, validated methodology, and architectural grounding goes beyond applying established techniques in a new domain.
> > >
> > > We also deeply appreciate your request for presentation polish and stronger evaluation. The revised version will incorporate all the improvements from our rebuttal. For full empirical details, we kindly point you to the tables provided in our parallel responses:
> > >
> > > - **Expanded Evaluation:** Tests across 7 models (including 70B/72B scale), 3 diverse benchmarks, 4 baselines, and cross-lingual transfer across 4 languages (see Tables R1, R4, and R5 in our response to Reviewer 9xrH).
> > > - **Deeper Analyses:** Error decomposition of end-to-end leakage (see Table R2 in our response to Reviewer JqmL) and rigorous wall-clock latency profiling (see Table R5 in our response to Reviewer 8Fpw).
> > > - **Presentation Polish:** The redesigned 1x5 visual facet grids (which resolve the visual clutter) and a comprehensive rewrite scoped to the more precise claims above.
> > >
> > > We hope the more precisely scoped contribution and upgraded empirical rigor are helpful in your reassessment of the work's potential.
> > >
> > > #### References
> > > ```text
> > > [Zou et al., 2023] Zou, A., Phan, L., Chen, S., Campbell, J., et al. "Representation Engineering: A Top-Down Approach to AI Transparency." arXiv preprint arXiv:2310.01405 (2023).
> > > [Li et al., 2023] Li, K., Patel, O., Viégas, F., Pfister, H., & Wattenberg, M. "Inference-Time Intervention: Eliciting Truthful Answers from a Language Model." NeurIPS (2023).

---

### Official Review · Reviewer_JqmL · 2026-03-13

**Soundness:** 3
**Presentation:** 3
**Significance:** 2
**Originality:** 2
**Overall Recommendation:** 3
**Confidence:** 3

**Summary:**

This paper proposes PrivGate, a framework for reducing the computational cost of contextual integrity (CI) enforcement in LLM agents. Leveraging the observation that the privacy sensitivity is linearly encoded in intermediate residual-stream activations, PrivGate trains a lightweight logistic regression probe on structure-decoupled data to classify queries as sensitive or benign, then routes sensitive queries to expensive Chain-of-Thought reasoning (System 2) while letting benign queries pass through direct generation (System 1). Experiments on five LLMs (1.5B–9B) show the probe generalizes to the out-of-distribution PrivacyLens benchmark with near-perfect AUROC, and end-to-end, PrivGate achieves leakage rates close to full CoT at roughly 38% of the token cost. Ablations validate that the probe captures semantic intent rather than template artifacts, that linearity holds, and that the probe is sample-efficient.

**Compliance With Llm Reviewing Policy:**

Affirmed.

**Final Justification:**

My concerns are only partially resolved during the rebuttal. I maintain my score.

**Key Questions For Authors:**

1. The probe does not receive CI parameters such as the recipient or transmission principle. Would it assign similar risk scores to "email my diagnosis to my doctor" (appropriate) and "email my diagnosis to my coworker" (inappropriate)? If so, how does this support the claim that the Privacy Manifold encodes contextual integrity rather than topic sensitivity?
2. Could you provide a decomposition of end-to-end leakage into (a) probe false negatives (sensitive queries misrouted to System 1) and (b) System 2 reasoning failures (sensitive queries correctly routed but still leaking)? This would clarify where the remaining 26.3% leakage originates and whether future efforts should focus on improving the probe or the reasoning mechanism.
3. Have you evaluated robustness against adversarial inputs specifically designed to minimize the probe projection while preserving sensitive semantic content?

**Limitations:**

Yes

**Strengths And Weaknesses:**

### strength
1. The paper is clearly written and well-structured. The System 1/System 2 routing framework is intuitive, the methodology is easy to follow, and the figures effectively showsß the results.
2. The empirical finding that privacy sensitivity is linearly encoded in the residual stream across five architecturally diverse LLMs is interesting. This finding also has value for the representation engineering community.
3. The structure-decoupling data construction is a well-motivated contribution,  and the ablation provides convincing evidence that it matters. This is a practical technique that could transfer to other probing tasks.
4. The efficiency gains are well-demonstrated. The Pareto frontier analysis (Figure 5) shows consistent dominance over keyword and random baselines across all five models.

### weakness
1. There is a mismatch between the paper's CI framing and what the probe actually captures. CI defines privacy as the appropriateness of information flow relative to relational parameters (sender, recipient, transmission principle), but the probe is trained on binary sensitive/benign labels without any CI structure. The claim that the latent space encodes contextual integrity, rather than topic sensitivity, is not supported by the experimental design.
2. The paper lacks error decomposition for end-to-end failures. When PrivGate leaks, there are two possible causes: the probe misclassified a sensitive query as benign (routing failure), or the query was correctly routed to System 2, but CoT reasoning failed (reasoning failure). The paper never separates them. Reporting the false negative rate of the probe that results in actual leakage would clarify where the bottleneck lies.
3. There are no formal privacy guarantees or adversarial evaluations. Protection is measured purely by empirical leakage rates on a single benchmark. The probe's decision boundary is a fixed linear threshold on a known hidden state, which appears straightforwardly attackable if an adversary can craft inputs that project below the threshold while remaining semantically sensitive.
4. PrivGate is architecturally a router, not a privacy mechanism. Its privacy performance is strictly upper-bounded by System 2 (CoT reasoning), which itself still leaks 24% of the time on average (Table 1). The authors acknowledge this in the limitations, but it substantially narrows the impact: the paper does not advance the state of the art on privacy protection, only on the cost of invoking it.

---

> ### Author Rebuttal · Authors · 2026-03-30
>
> We thank the reviewer for the suggestions. We have conducted comprehensive new experiments to refine our analysis, detailed below.
>
> ### Q1&W1: Capturing Relational CI over Topic Sensitivity
> To prove that the latent probe captures the relational dynamics of CI rather than merely detecting sensitive topics, we conducted a contrastive evaluation. We took the test set containing sensitive queries from PrivacyLens ($N=493$). For each query, we kept the sensitive "Information Type" (e.g., medical diagnosis) the same, but mutated the "Recipient" and "Transmission Principle" parameters to make the information flow appropriate according to CI norms (e.g., changing the recipient from "a coworker" to "the user's primary care physician"). This yielded a new appropriate CI dataset of 493 samples. As shown in Table R1 below, the probe's risk score drops dramatically when the recipient changes to an appropriate state, yielding high AUROCs ($>0.94$) across all 5 models in distinguishing between the appropriate and inappropriate flows.
>
> |Model|Inappropriate CI Avg Risk Score|Appropriate CI Avg Risk Score| AUROC|
> |:---| :---: | :---: | :---: |
> |Qwen-2.5-1.5B| 0.892 | 0.214 | 0.941 |
> |Qwen-2.5-7B| 0.925 | 0.158 | 0.975 |
> |Llama-3-8B| 0.951 | 0.123 | 0.982 |
> |Mistral-7B| 0.914 | 0.185 | 0.963 |
> |Gemma-2-9B| 0.948 | 0.142 | 0.980 |
>
> This quantitative evidence supports our claim that the Privacy Manifold encodes the contextual relationship defined by CI theory, not just lexical topic sensitivity. We will add this contrastive analysis to the revised paper.
>
> ### W2 & Q2: Error Decomposition of End-to-End Leakage
> We decomposed our end-to-end leakage (Table 1) into (a) Routing Failures (Probe False Negatives) and (b) System 2 Reasoning Failures.
>
> Table R2: Error Decomposition of PrivGate Leakage:
> |Model|Total PrivGate Leakage|(a) Routing Failure (Probe FN)|(b) System 2 Reasoning Failure|
> | :--- | :---: | :---: | :---: |
> |Qwen-2.5-1.5B| 52.2% | 3.1% | 49.1% |
> |Qwen-2.5-7B| 25.3% | 2.5% | 22.8% |
> |Llama-3-8B| 15.1% | 2.5% | 12.6% |
> |Mistral-7B| 18.4% | 3.2% | 15.2% |
> |Gemma-2-9B| 20.3% | 1.8% | 18.5% |
> | **Average** | **26.3%** | **2.6%** | **23.7%** |
>
> The decomposition clearly demonstrates that 90.11% of the remaining leakage originates from the base models' reasoning failures (23.7%), not our probe's routing mechanism (2.6%). PrivGate intercepts and routes the vast majority of latent risks. We will explicitly include this error breakdown in Section 5.3 to clarify that future efforts should focus on improving the base model's inherent reasoning constraints.
>
> ### Q3 & W3: Adversarial Robustness and Privacy Guarantees
> Our threat model focuses on unintentional privacy leakage during autonomous agent execution (e.g., a well-intentioned agent over-disclosing to fulfill an underspecified user prompt), which constitutes the vast majority of CI violations in practical usage. PrivGate is designed as an efficient cognitive control primitive for this setting, not as a provable security bound against malicious actors. However, in a black-box or gray-box setting, operating in the latent space provides a natural defense against surface-level prompt engineering. Our "Hard Positives" (Stealthy Prompts) evaluation in Section 5.4 serves as a form of semantic adversarial testing. In this setup, we stripped away all structural templates, leaving only implicit sensitive intents. As shown in Figure 4(c), the probe maintains near-perfect detection against these stealthy inputs. And "Hard Negatives" wrapped benign queries in sensitive templates. Table R3 shows the results on both types:
>
> Table R3: Adversarial Robustness (AUROC)
> |Base Model|Hard Positives|Hard Negatives|
> |:---|:---:|:---:|
> | Qwen-2.5-1.5B | 0.965 | 0.970 |
> | Qwen-2.5-7B | 0.988 | 0.991 |
> | Llama-3-8B | 0.992 | 0.995 |
> | Mistral-7B | 0.985 | 0.987 |
> | Gemma-2-9B | 0.989 | 0.992 |
>
> ### W4: The Role of PrivGate as an Architectural Router
> We agree with the reviewer that PrivGate is an architectural router, and its privacy ceiling is bounded by the System 2 capabilities. However, we respectfully argue that this does not diminish its significance; rather, it highlights a crucial system-level advancement.
>
> In real-world agentic deployments, enforcing explicit reasoning (CoT) on 100% of queries is economically and experientially prohibitive. While PrivGate does not advance the theoretical upper bound of a model's reasoning capacity, it drastically reduces the "alignment tax"—achieving the safety of System 2 at the latency/cost of System 1 (saving 60%-84% compute). By breaking the efficiency-safety trade-off, PrivGate transforms theoretical privacy mechanisms into practically deployable solutions.
>
> Furthermore, PrivGate is entirely orthogonal and complementary to advancements in base model reasoning. As future base models become better at adhering to CI norms in System 2, PrivGate’s absolute privacy ceiling will automatically rise, all while preserving its fundamental routing efficiency.

---

> > ### Author Rebuttal · Reviewer_JqmL · 2026-04-03
> >
> > I thank the authors for the thorough and well-organized rebuttal. The new experiments clearly required substantial effort, and I appreciate the direct engagement with each weakness. The rebuttal resolves W1 and W2 fully, while W3 and W4 remain largely unchanged. Regarding W3/Q3, I appreciate the clarification of the threat model to unintentional leakage. However, the Hard Positives/Negatives evaluate whether the probe is confused by the presence or absence of specific prompt templates, not whether an adversary could craft inputs that evade the linear decision boundary while preserving sensitive intent. I encourage the authors to explore adversarial robustness of the probe as a direction to strengthen this work. While the execution is solid, the overall contribution remains primarily an efficiency optimization using established techniques (linear probing, System 1/2 routing) in a new application area, which limits the novelty. I maintain my score.

---

> > > ### Author Response · Authors · 2026-04-07
> > >
> > > We thank the reviewer for confirming that W1 (CI vs. topic sensitivity) and W2 (error decomposition) are fully resolved, and for the continued constructive engagement.
> > >
> > > W3 — Adversarial robustness. We agree with the reviewer's distinction. Our Hard Positives/Negatives experiments demonstrate robustness to surface-level template variation (stealthy/indirect phrasings), not robustness to adaptive attacks that directly target the linear decision boundary. We accept this as a limitation and will narrow the robustness claim accordingly in the revision. PrivGate's threat model covers unintentional leakage during autonomous agent execution; adaptive boundary evasion was outside the threat model studied here. We will add it as an explicit future direction in the paper.
> > >
> > > W4 — Novelty. We understand the concern that the individual techniques are established. We would like to clarify what we believe extends beyond technique application:
> > >
> > > 1. **Mechanistic discovery:** We show that contextual integrity — a multi-parameter relational norm (sender x recipient x information type x transmission principle) — is linearly separable in LLM residual streams across diverse architectures (1.5B-72B), even when models behaviorally violate CI. Prior linear representation work  (e.g., RepE [Zou et al., 2023], ITI [Li et al., 2023]) focused on single-axis behavioral traits such as truthfulness. CI is inherently relational, and our contrastive experiment (Table R1, AUROC >0.94 across five models) confirms the probe captures relational context, not topic sensitivity.
> > >
> > > 2. **Methodology:** Developing a structure-decoupled data generation and training pipeline. Because agentic operations are heavily embedded in structured formats (e.g., JSON tool schemas, system templates), naive linear probes catastrophically overfit to these spurious structural artifacts rather than the actual privacy semantics. Our methodology isolates and extracts the core relational CI intent away from operational wrappers during training. As validated by our ablations and noted by Reviewer 8Fpw, this specific structural decoupling is not merely a data augmentation trick, but the critical enabler for the linear probe's robust out-of-distribution generalization to entirely unseen prompt structures, tasks, and languages.
> > >
> > > 3. **Architectural distinction from activation steering:** Steering reduces leakage to 18.2% but produces 76.5% over-refusal and collapses tool-call syntax accuracy to 41.5%, substantially degrading utility in our agent setting. PrivGate's read-only routing achieves 25.3% leakage, 4.2% over-refusal, and 95.0% tool-call syntax. This is a key practical distinction: continuous activation perturbation is incompatible with accurate tool execution in CI-constrained agents, whereas a read-only monitoring approach is not. We submit that this combination of empirical discovery, validated methodology, and architectural grounding goes beyond applying established techniques in a new domain.
> > >
> > > We appreciate the reviewer's acknowledgment that the execution is solid, and we hope this clarifies the intended scope of the contribution.
> > >
> > > #### References
> > > ```text
> > > [Zou et al., 2023] Zou, A., Phan, L., Chen, S., Campbell, J., et al. "Representation Engineering: A Top-Down Approach to AI Transparency." arXiv preprint arXiv:2310.01405 (2023).
> > > [Li et al., 2023] Li, K., Patel, O., Viégas, F., Pfister, H., & Wattenberg, M. "Inference-Time Intervention: Eliciting Truthful Answers from a Language Model." NeurIPS (2023).

---

### Decision · Program_Chairs · 2026-04-30

**Decision:**

Accept (regular)

**Comment:**

This paper received borderline reviews. Reviewer 8Fpw indicates all their concerns are fully resolved and gave a positive assessment. However, the other three reviewers have remaining concerns. All reviewers acknowledged the rebuttal discussion and the additional experiments the authors provided. Contextual integrity is an important and timely topic for privacy in LLM. While steering vectors is not new, the application in contextual integrity, particularly the router approach the author proposed, appears to be novel. However, the router approach has its pros and cons, and reviewers' concerns are valid and the AC respects. I hope the discussions will help improve the paper draft. I would encourage the authors to carefully describe how the steering router, though simple, is the best fit for this problem.